# Gearbox Fault Diagnosis Based on Refined Time-Shift Multiscale Reverse Dispersion Entropy and Optimised Support Vector Machine

**Xiang Wang** [1,*] **and Han Jiang** [2]

1    School of Energy and Power Engineering, Nanjing Institute of Technology, Nanjing 211167, China
2    School of Electrical Engineering, Nanjing Institute of Technology, Nanjing 211167, China; y00450210821@njit.edu.cn
\*    Correspondence: wangxiang@njit.edu.cn

**Abstract:** The fault diagnosis of a gearbox is crucial to ensure its safe operation. Entropy has become a common tool for measuring the complexity of time series. However, entropy bias may occur when the data are not long enough or the scale becomes larger. This paper proposes a gearbox fault diagnosis method based on Refined Time-Shifted Multiscale Reverse Dispersion Entropy (RTSMRDE), t-distributed Stochastic Neighbour Embedding (t-SNE), and the Sparrow Search Algorithm Support Vector Machine (SSA-SVM). First, the proposed RTSMRDE was used to calculate the multiscale fault features. By incorporating the refined time-shift method into Multiscale Reverse Dispersion Entropy (MRDE), errors that arose during the processing of complex time series could be effectively reduced. Second, the t-SNE algorithm was utilized to extract sensitive features from the multiscale, high-dimensional fault features. Finally, the low-dimensional feature matrix was input into SSA-SVM for fault diagnosis. Two gearbox experiments showed that the diagnostic model proposed in this paper had an accuracy rate of 100%, and the proposed model performed better than other methods in terms of diagnostic performance.

**Keywords:** data reduction; fault diagnosis; gearbox; reverse dispersion entropy; support vector machine

## 1. Introduction

The gearbox is an important component of mechanical equipment [1]. Continuous work under complex conditions can easily lead to equipment damage [2]. When the gearbox malfunctions, the entire operating equipment will have safety hazards, and the fault signals will exhibit complexity and autocorrelation [3,4]. The implementation of safe and reliable fault diagnosis techniques can effectively prevent serious failures, reduce the operating and maintenance costs, and improve the reliability on mechanical equipment [5].

Due to the highly nonlinear and nonstationary nature of vibration signals from gearbox faults, linear feature extraction methods are not applicable [6]. Therefore, based on nonlinear dynamic theory, entropy methods have been widely used in the detection of rotating machinery dynamic behaviour [7,8]. Entropy, as a parameter for evaluating the regularization process and complexity of time series, is particularly suitable for characterizing nonstationary and nonlinear acceleration signals [9]. Various entropy methods, such as approximate entropy [10], sample entropy [11], fuzzy entropy [12], and permutation entropy [13], have been proposed and used in fault diagnosis. These methods, however, only consider signals at one scale and could ignore crucial temporal information. To overcome this limitation, Costa et al. [14] proposed multiscale entropy. Wang et al. [15] constructed statistical features based on multiscale sample entropy (MSE) to reflect the information of rotating machinery. Zheng et al. [16] combined multiscale fuzzy entropy (MFE) with the SVM to construct an intelligent diagnosis model. Chen et al. [17] combined local mean decomposition and multiscale permutation entropy (MPE) to enhance the feature extraction

ability of MPE through pre-processing of the initial signal. These methods have their own deficiencies. Specifically, MSE cannot measure the similarity of short time series, and the calculation of MSE and MFE is very time-consuming for long time series. MPE also ignores the difference between adjacent amplitude values in the time series, which may lead to the neglect of useful information in the amplitude.

Dispersion Entropy (DE) [18] is a new nonlinear dynamic index that overcomes the drawbacks of other entropy algorithms in measuring the complexity and irregularity of time series. When calculating DE, the amplitude of the time series is considered. However, the recognition ability of DE is not very good when dealing with autocorrelated signals. To solve this problem, Azami and Escudero [19] proposed the Dispersion Entropy algorithm based on multiscale fluctuation, which has demonstrated its stability and good recognition ability in processing signals of neurological disorders. Li et al. [20] proposed the reverse dispersion entropy (RDE) algorithm to detect sensor signals and demonstrated its excellent stability through testing with real ship signals. Meanwhile, Xing et al. applied the multiscale method to the RDE algorithm and achieved good results [21]. However, there are still some problems with the multiscale method [22]. Coarsening the time series can lead to the loss of important information. As the scale factor increases, the coarse-grained sequence becomes shorter, leading to greater entropy value bias [23]. Therefore, an improved MRDE algorithm needs to be proposed to overcome the above shortcomings by improving the existing coarse-graining method.

Fault diagnosis usually requires extracting a large number of fault feature sets from signals, which have characteristics such as nonlinearity and high dimensionality. Although these features can provide useful fault information, there may be many redundant features mixed in, which do not help improve the classification accuracy of the classifier but instead increase the burden on the classifier and reduce the classification efficiency. Therefore, appropriate dimensionality reduction algorithms are needed to reduce the dimensionality of the feature set. Classic data dimensionality reduction methods such as Principal Component Analysis (PCA) [24], Linear Discriminant Analysis (LDA) [25], Locally Linear Embedding (LLE) [26], t-Distributed Stochastic Neighbour Embedding (t-SNE) [27], and Isometric Mapping (Iso-map) [28] have been widely used in many fields. However, PCA and LDA are linear transformation methods for dimensionality reduction, which may produce significant bias when analysing nonlinear samples. Notably, t-SNE is a non-linear dimensionality reduction method for manifold learning, which performs better in preserving the local structure of data than other manifold learning methods. Using t-SNE to map high-dimensional data to low-dimensional space facilitates observing the distribution of data. In addition, the obtained low-dimensional sensitive feature set can improve classification efficiency and reduce storage requirements [29].

In the past few years, various classifiers have been applied in fault diagnosis, such as the k-Nearest Neighbour (k-NN) [30], Random Forest (RF) [31], Artificial Neural Network (ANN) [32], and Support Vector Machine (SVM) [33], etc. In small-sample applications, SVM has higher generalization ability and more accurate classification results. It is worth noting that suitable parameters have a significant impact on SVM performance. Chen et al. [34] used the Particle Swarm Optimization (PSO) to automatically search for the best parameters of SVM. Dong et al. [13] used the Grey Wolf Optimization (GWO) to achieve good results. However, many optimization algorithms exhibit poor local search capabilities and slow convergence speeds when solving complex problems. The Sparrow Search Algorithm [35] (SSA) simulates the foraging and anti-predator behaviour of sparrows and establishes a mathematical model by flexibly using producer and scavenger strategies. The SSA has the characteristics of fast convergence speed and strong optimization ability. This paper introduces the SSA to select the best parameters for SVM.

This paper proposes a novel feature extraction method based on RTSMRDE. First, RDE was used instead of traditional DE. Second, inspired by the time-shift process, a refined time-shift coarse-grained approach was used to reconstruct sub-sequences. Finally, an improved algorithm was used to extract fault features and reduce errors. The

extracted fault feature matrix was then dimensionally reduced using t-SNE, resulting in a sensitive low-dimensional feature matrix. This feature matrix was input into SSA-SVM for fault classification. Simulation signals and experimental analysis were used to verify the effectiveness.

Summarizing the above analysis, the main contributions are as follows:

1. This paper proposes a novel RTSMRDE method for the multiscale feature extraction of gearbox faults.
2. Utilizing data dimensionality reduction methods to extract sensitive features from the initial high-dimensional feature matrix, resulting in more accurate fault recognition.
3. Constructing an intelligent diagnosis model for gearbox based on RTSMRDE, t-SNE, and SSA-SVM.
4. Validating the effectiveness through simulation signals, gearbox datasets, and experimental data. The experimental results indicate that the fault diagnosis model performs significantly better than six other methods in terms of overall performance.

The rest of this paper is structured as follows: Section 2 introduces RTSMRDE algorithms and selects the optimal parameters of the algorithms through experiments. Section 3 overviews t-SNE and SSA-SVM and proposes the fault diagnosis model. Section 4 verifies the fault diagnosis method through two experiments. Finally, Section 5 provides a brief summary of the work in this paper.

## 2. Refined Time-Shift Multiscale Reverse Dispersion Entropy

### 2.1. Reverse Dispersion Entropy

Based on the theory of PE, RDE is a new time-series complexity analysis method that combines the positive aspects of DE and Reverse Permutation Entropy. The following is a description of the RDE steps [20]:

Step 1. Assuming we have a univariate signal of length $X = \{x_1, x_2, \cdots, x_N\}$, $X$ is mapped into $Y = \{y_1, y_2, \cdots, y_N\}$ by the standard normal cumulative distribution function (NCDF).

$$y_i = \frac{1}{\sigma\sqrt{2\pi}} \int_{-\infty}^{x_i} e^{\frac{-(t-\gamma)^2}{2\sigma^2}} dt \qquad (1)$$

where $y_i \in (0, 1)$; $\sigma$ and $\gamma$ denote the standard deviation and mean of $X$, respectively.

Step 2. Mapping $Y$ to $c$ classes. We map $Y$ to $Z^c = \{z_1^c, z_2^c, \cdots, z_N^c\}$ using $round(c \cdot y_i + 0.5)$, where $c$ is the number of classes, and $z_i$ is a positive integer from 1 to $c$.

Step 3. $Z$ is reconstructed to embedding vectors $T$ using time delay $d$ and embedding dimension $m$, respectively. The matrix comprising embedding vectors can be expressed as follows:

$$\begin{bmatrix} \left\{ z_1^c, z_{1+d}^c, \cdots, z_{1+(m-1)d}^c \right\} \\ \vdots \qquad \vdots \\ \left\{ z_j^c, z_{j+d}^c, \cdots, z_{j+(m-1)d}^c \right\} \\ \vdots \qquad \vdots \\ \left\{ z_t^c, z_{t+d}^c, \cdots, z_{t+(m-1)d}^c \right\} \end{bmatrix} \qquad (2)$$

where $N - (m-1)d$ is the number of embedding vectors $t$.

Step 4. Each value $z_j$ in each vector group corresponds to the subscript of pattern $\pi_{v_0 v_1 \cdots v_{m-1}}$, as shown below:

$$z_j^c = v_0, z_{j+d}^c = v_1, z_{j+2d}^c = v_2, \cdots, z_{j+(m-1)d}^c = v_{m-1} \qquad (3)$$

where $c^m$ is the number of potential dispersion patterns, because each $z_j^c$ has $m$ members, and each one can be an integer between 1 and $c$.

Step 5. The following is an expression for the *i*th dispersion pattern's relative frequency:

$$p(\pi_i) = \frac{Number\{\pi_i\}}{N - (m-1)d}(1 \leq i \leq c^m) \tag{4}$$

where $p(\pi_i)$ represents the probability of the *i*th dispersion patterns.

Step 6. The RDE calculation illustrated below:

$$RDE(x, m, c, d) = \sum_{i=1}^{c^m} \left( p(\pi_i) - \frac{1}{c^m} \right)^2 \tag{5}$$

When $p(\pi_i) = 1/c^m$, the value of $RDE(x, m, c, d)$ is 0 (minimum value). When $p(\pi_i) = 1$, $\pi_i$ is the only existing dispersion pattern, the value of $RDE(x, m, c, d)$ is $1 - 1/c^m$ (maximum value). Thus, the normalized RDE can be expressed as follows:

$$NRDE = \frac{RDE(x, m, c, d)}{1 - 1/c^m} \tag{6}$$

### 2.2. Multiscale Reverse Dispersion Entropy

MRED is an improved algorithm combining RDE with multiscale entropy. Non-overlapping multiscale means are calculated at different scales to form a new sequence, and then the RDE values are calculated. The MRDE algorithm can be described below.

For time series $X = \{x_1, x_2, \cdots, x_N\}$, the initial time series $X$ is divided into non-overlapping segments of length $s$. Then, the mean of each segment is calculated, and they are arranged in order together. This process is called coarse graining, as shown below:

$$y_j^s = \frac{1}{s} \sum_{i=(j-1)s+1}^{js} x_i, 1 \leq j \leq \lfloor L/s \rfloor \tag{7}$$

where $s$ is scale factor. When $s = 1$, $y^1$ is the initial time series. When $s > 1$, the original sequence is divided into $s$ coarse graining sequences of length $\lfloor L/s \rfloor$.

For each set of coarse graining data, RDE is calculated as:

$$MRDE(X, m, c, d, s) = RDE(y^s, m, c, d) \tag{8}$$

MRDE analysis is used to calculate the RDE of the coarse-grained sequence under multiple scales. The method overcomes the limitation of RDE in measuring the complexity of signals only at a single scale. MRDE uses a coarse-grained multiscale process, but it has a high requirement for the length of the time series. When the scale factor increases, the length of the coarse-grained sequence becomes shorter, which may result in larger errors. In other words, during the coarse-graining process, the original signal is blurred into multiple signals, which simplifies the calculation but also poses a risk of ignoring important fault information.

### 2.3. Refined Time-Shift Multiscale Reverse Dispersion Entropy

In this subsection, RTSMRDE is proposed aiming at the above problems of MRDE, and the detailed steps of RTSMRDE can be described as follows:

For time series $X = \{x_1, x_2, \cdots, x_N\}$, $\beta$ and $\alpha$ are positive integers, where $\beta = 1, 2, \cdots, \alpha$; then, $\alpha$ new time series can be constructed by:

$$u_\beta^\alpha = \left\{ x_\beta, x_{\beta+\alpha}, x_{\beta+2\alpha}, \cdots, x_{\beta+\lfloor(N-\beta)/\alpha\rfloor\alpha} \right\} \tag{9}$$

where $\alpha$ is the time-shift interval; $\lfloor(N-\beta)/\alpha\rfloor$ represents positive integers less than or equal to $(N-\beta)/\alpha$.

For a given scale factor $\alpha$, $u_\beta^\alpha$ is composed of $(N/\alpha)$ signal points. Thus, the $p_\beta^\alpha\left(\pi_{v_0 v_1 \cdots v_{m-1}}\right)$ of each $u_\beta^\alpha$ can be acquired using RDE. Then, we define the probability average of time-shift multiscale coarse-grained sequences when scale equal to $\alpha$.

$$\overline{p_\alpha}\left(\pi_{v_0 v_1 \cdots v_{m-1}}\right) = \frac{1}{\alpha} \sum_{\beta=1}^{\alpha} p_\beta^\alpha\left(\pi_{v_0 v_1 \cdots v_{m-1}}\right) \tag{10}$$

The final RTSMRDE value at scale $\alpha$ as follow:

$$RTSMRDE(X, m, c, d, \alpha) = \sum_{i=1}^{c^m} \left(\overline{p_\beta^\alpha}\left(\pi_{v_0 v_1 \cdots v_{m-1}}\right) - \frac{1}{c^m}\right)^2 \tag{11}$$

The RTSMRDE algorithm processes the original sequence using time-shift methods to form a sequence group consisting of multiple sub-sequences, each of which is still a part of the original sequence. This maximally preserves the information in the original signal, and the mean probability of the dispersion patterns also resolves the error caused by the increasing scale factor. The flowchart of RTSMRDE is shown as Figure 1.

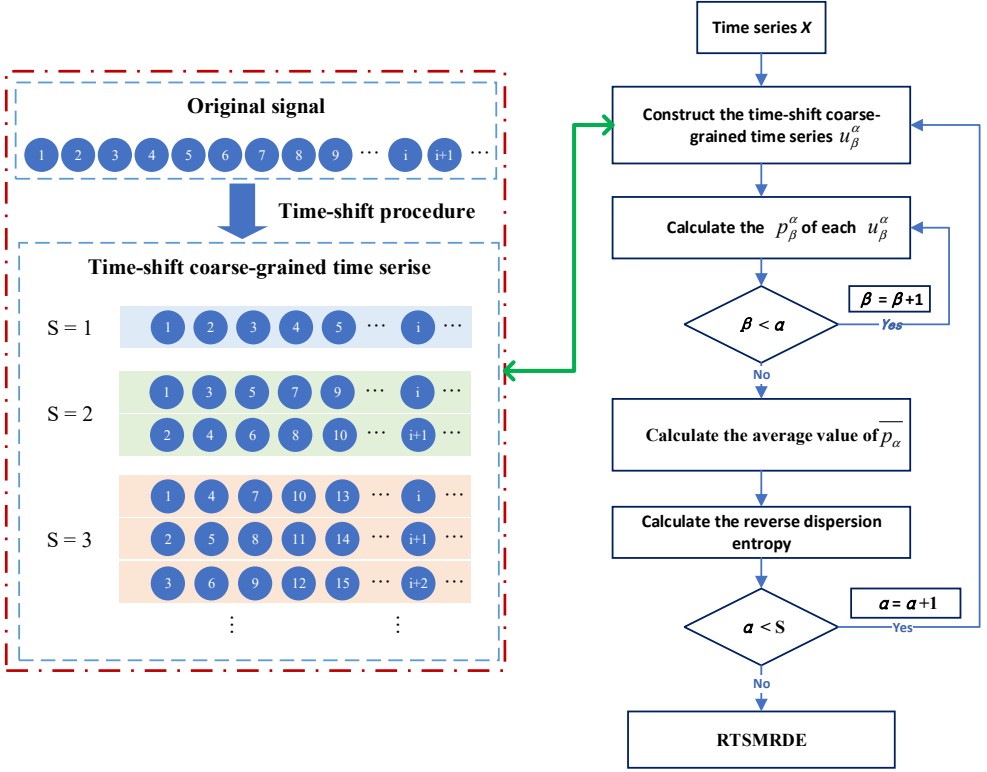

**Figure 1.** The flowchart of the RTSMRDE method.

### 2.4. Parameters Selection

There are four parameters for the RTSMRDE that must be manually configured, namely the scale *s*, the time delay *d*, the embedding dimension *m*, and the class *c*. In total, 30 sets of pink noise (pn) and white noise (wn) were used as test signals. In machine fault detection, white noise refers to a random noise signal with uniform power spectral density whose energy in the frequency domain is basically equal in each frequency band and has no obvious frequency distribution characteristics; pink noise is a type of noise signal with a $1/f$ power spectral density distribution characteristic, and the energy in its low-frequency part is more than that in the high-frequency part.

First, we considered how the algorithm would be affected by the time delay *d* and scaling factor *s*. According to [20], there may be confusion when the time delay *d* exceeds 1,

even though the impact of time delay on the algorithm is minimal. Therefore, we decided to set the time delay to 1. In addition, choosing an incorrect scale factor *s* could pose a challenge in accurately extracting the signal's fault feature information. If the scale factor is too large, it can negatively impact the algorithm's performance and generate inaccurate entropy values. In order to obtain reliable results, we chose to comprehensively consider a scale factor of 20.

Second, we considered the impact of the embedding dimension *m* on the RTSMRDE algorithm. Regarding the selection of the embedding dimension *m*, if it is set too small, it may be difficult to detect the dynamic behaviour of the signal. As shown in Figure 2, the entropy values of both the signal and the noise decrease with increasing *m*, but the general pattern remains the same. However, as shown in Table 1, increasing *m* will require more processing time. Moreover, the entropy values between different noises tend to converge at larger scales. Therefore, we decided to set *m* equal to 2.

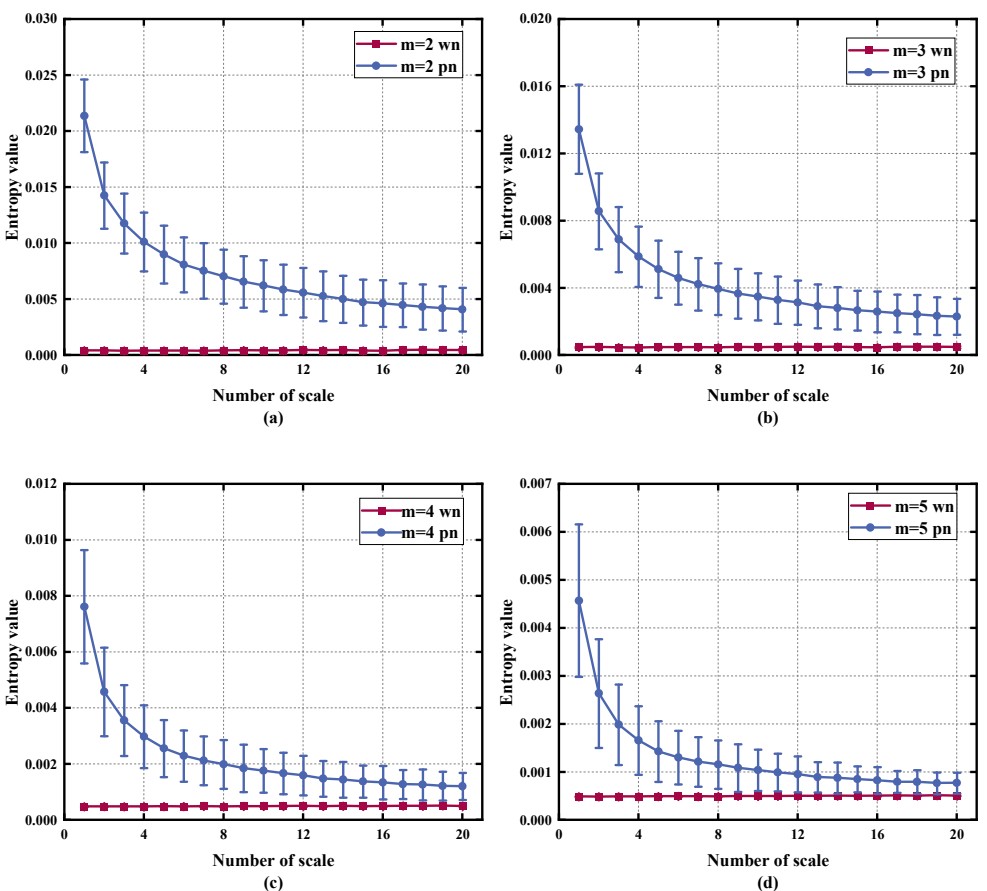

**Figure 2.** Different *m* on RTSMRDE: (**a**) *m* = 2, (**b**) *m* = 3, (**c**) *m* = 4, (**d**) *m* = 5.

**Table 1.** Running time of RTSMRDE for different *m* values.

| Type | *m* = 2 | *m* = 3 | *m* = 4 | *m* = 5 |
|---|---|---|---|---|
| Seconds | 1.121 s | 3.403 s | 14.575 s | 80.805 s |

Finally, we considered the impact of varying values of *c* on the algorithm, with *m* set to 2. If *c* is set too small, it will be challenging to distinguish between classes with varying amplitudes. On the other hand, a large value of *c* can increase the system's susceptibility to noise and result in a higher computational load. Figure 3 shows that, despite a decrease in the entropy values of the two noises with higher *c* values, the overall trend remained largely unchanged. However, setting too many categories can significantly increase the

computation time, as shown in Table 2. Therefore, to strike a balance between reliable statistical measurements and computational efficiency, we decided to set the number of categories *c* equal to 6.

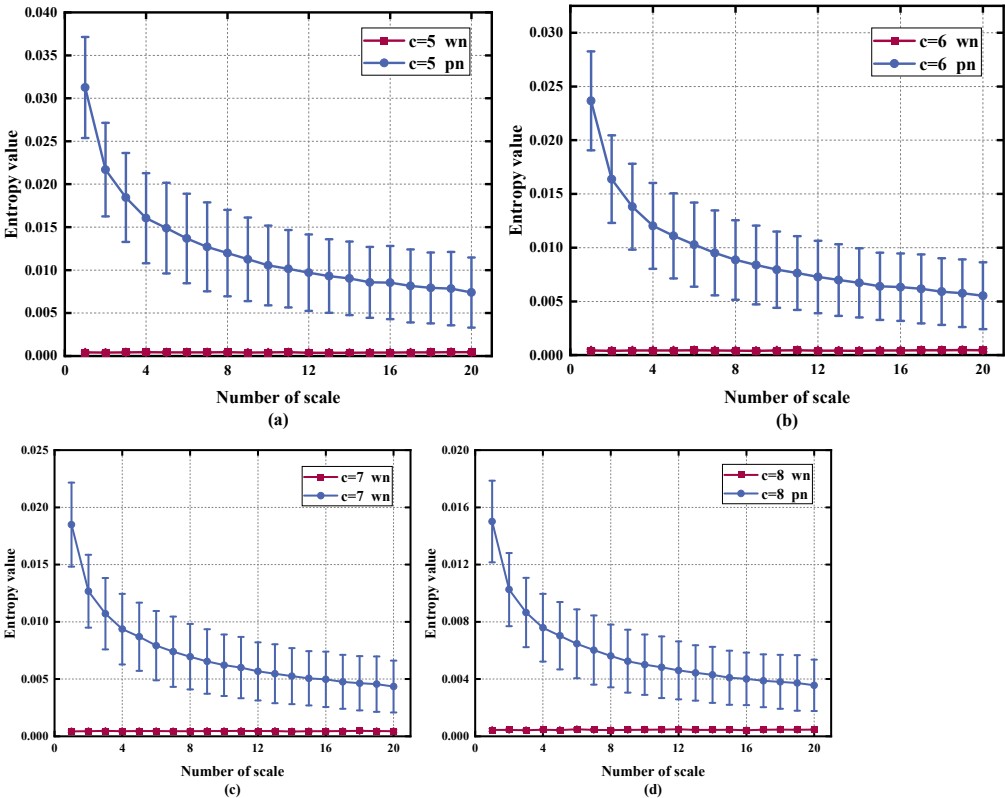

**Figure 3.** Different *c* on RTSMRDE: (**a**) *c* = 5, (**b**) *c* = 6, (**c**) *c* = 7, (**d**) *c* = 8.

**Table 2.** Running time of RTSMRDE for different *c*.

| Type | *c* = 5 | *c* = 6 | *c* = 7 | *c* = 8 |
|---|---|---|---|---|
| Seconds | 1.025 s | 1.092 s | 1.290 s | 1.549 s |

In conclusion, the appropriate values for the RTSMRDE parameters were m = 2, c = 6, d = 1, and s = 20.

### 2.5. Comparison of RTSMRDE and Other Entropy Methods Using White Noise and Pink Noise

We compared our proposed method to other existing entropy algorithms, with specific parameters listed in Table 3. For the convenience of comparison, we set the scale factor of the seven algorithms to 20. White noise and pink noise, each configured with 30 samples, were used for the comparison. The running times and results of the seven entropy algorithms are presented in Table 4 and Figure 4, respectively.

**Table 3.** Selection of parameters with different entropy.

| Entropy Methods | Parameters |
|---|---|
| MSE [36] | $m = 2, n = 2, d = 1, s = 20, r = 0.15$ SD |
| MFE [37] | $m = 3, d = 1, s = 20, r = 0.15$ SD |
| MDE [38] | $m = 3, c = 6, d = 1, s = 20$ |
| RCMDE [39] | $m = 2, c = 9, d = 1, s = 20$ |
| MRDE [40] | $m = 3, c = 5, d = 1, s = 20$ |
| RCMRDE [41] | $m = 2, c = 5, d = 1, s = 20$ |
| RTSMRDE (proposed) | $m = 2, c = 6, d = 1, s = 20$ |

**Table 4.** Running time for different entropies.

| Type | RTSMRDE | RCMRDE | MRDE | RCMDE | MDE | MFE | MSE |
|---|---|---|---|---|---|---|---|
| Seconds | 1.081 s | 4.576 s | 0.550 s | 4.380 s | 0.549 s | 4.315 s | 3.305 s |

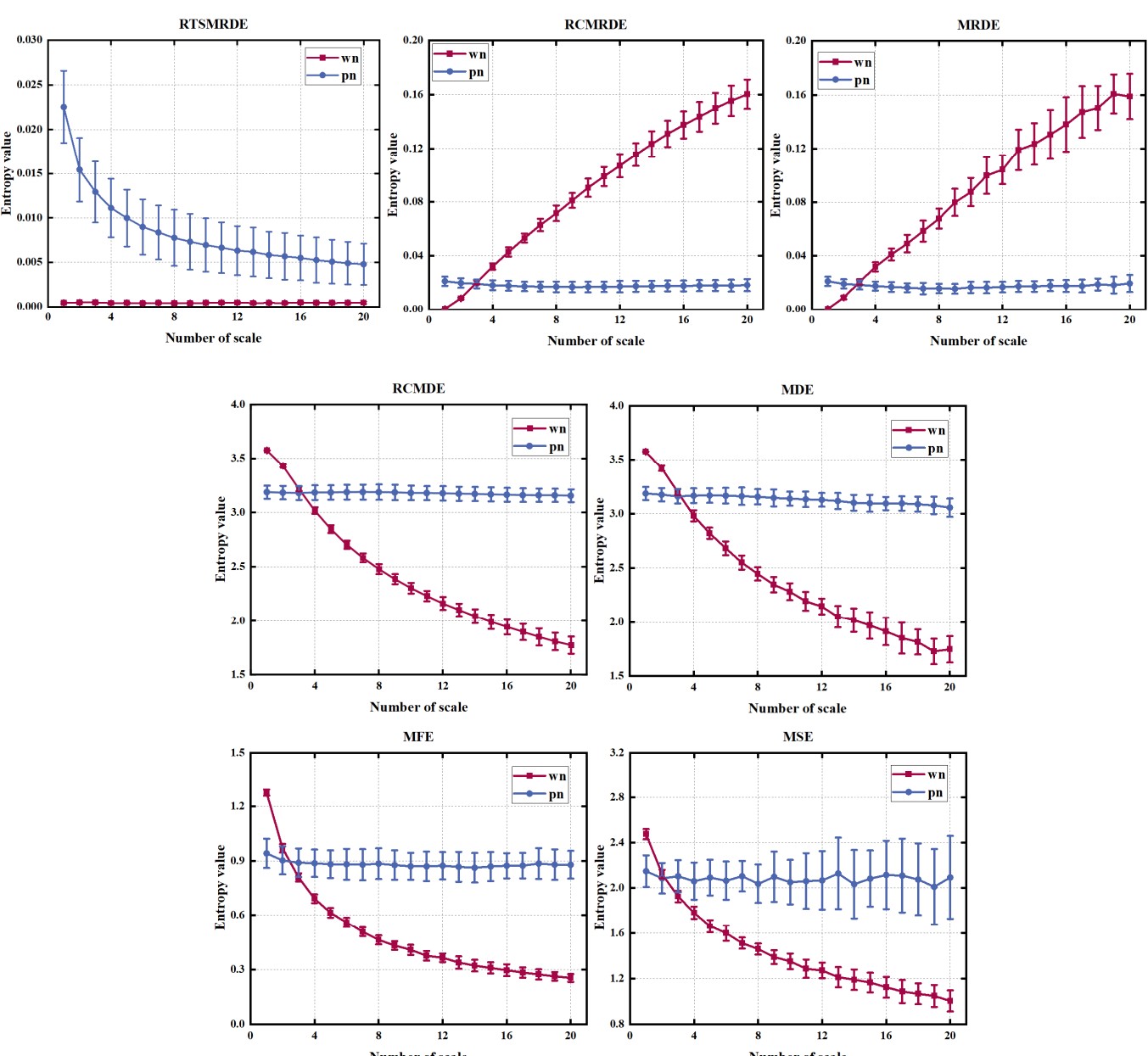

**Figure 4.** Different entropy values on white noise and pink noise.

As shown in Figure 4, the entropy curves calculated by MSE, MFE, MDE, and MRDE exhibited significant fluctuations with the increasing scale. Among them, MSE and MRDE also showed an increasing error at larger scale factors. In contrast, the entropy values derived by the entropy algorithm following refinement operation and composite coarse-grained structure were more stable. In addition, RTSMRDE combined with the time-shifted structure could correctly distinguish the two noises at full scale, and no crossover occurred. Furthermore, the reverse dispersion entropy was defined as the distance of the signal from the white noise, and only the white noise value of RTSMRDE was closest to 0 among RTSMRDE, RCMRDE, and MRDE, which was consistent with the reality. Finally, Table 4

shows that MDE and MRDE had the shortest calculation time. RTSMRDE had the next shortest time. The calculation time of the other methods was three to four times that of RTSMRDE. In conclusion, the RTSMRDE is of superior quality than the comparative algorithms mentioned above.

## 3. The Proposed Intelligent Gearbox Fault Diagnosis Method

### 3.1. Data Reduction Method

The t-SNE is a nonlinear dimensionality reduction algorithm commonly used for the dimensionality reduction and visualization of high-dimensional data. The algorithm maps high-dimensional data into low-dimensional space, so that the relative distance between data points can be preserved, and the effect of dimensionality reduction and visualization is achieved. Specifically, the t-SNE algorithm defines similarity matrices in high-dimensional and low-dimensional spaces, respectively, and optimizes the mapping relationship by minimizing the Kullback–Leibler divergence between the two similarity matrices. Through the gradient descent algorithm, each data point in the high-dimensional space is mapped to the corresponding data point in the low-dimensional space. Compared with the traditional linear dimensionality reduction algorithm, the t-SNE algorithm can preserve the local structure information between data more completely, so it performs better in visualizing high-dimensional data. More detailed algorithm steps can be found in the relevant literature [27].

### 3.2. Support Vector Machine

SVM is an efficient classification model. The basic idea of SVM is to find an optimal hyperplane in the feature space, which separates the samples of different classes and maximizes the margin between different samples. In SVM, the sample points closest to the hyperplane determine the location of the hyperplane. Those sample points are called support vectors. The ultimate goal of SVM optimization is to find the hyperplane with the largest margin and minimize the classification error. For non-linearly separable data sets, slack variables are usually introduced, or kernel functions are used to map the data into high-dimensional space. The specific algorithm of SVM can be referred to in [33].

### 3.3. Sparrow Search Algorithm

The SSA simulates the foraging and anti-predator behaviour of sparrows and establishes a mathematical model flexibly using producer and scavenger strategies. The specific theoretical method can be referred to in [35], and its mathematical model can be briefly described as follows.

Represent the position of sparrows using the following matrix:

$$M = \begin{bmatrix} M_{1,1} & M_{1,2} & \cdots & M_{1,y} \\ M_{2,1} & M_{2,2} & \cdots & M_{2,y} \\ \vdots & \vdots & \vdots & \vdots \\ M_{x,1} & M_{x,2} & \cdots & M_{x,y} \end{bmatrix} \tag{12}$$

where $x$ is the number of sparrows, and $y$ is the number of optimization parameters.

The fitness value of all sparrows can be expressed by matrix below:

$$F_M = \begin{bmatrix} f\left(\begin{bmatrix} M_{1,1} & M_{1,2} & \cdots & M_{1,y} \end{bmatrix}\right) \\ f\left(\begin{bmatrix} M_{2,1} & M_{2,2} & \cdots & M_{2,y} \end{bmatrix}\right) \\ \vdots \\ f\left(\begin{bmatrix} M_{x,1} & M_{x,2} & \cdots & M_{x,y} \end{bmatrix}\right) \end{bmatrix} \tag{13}$$

where $F_M$ represents the fitness value.

During the search process, producers with higher fitness have higher priority in obtaining food. The location update of the producer is described as:

$$M_{i,j}^{t+1} = \begin{cases} M_{i,j}^t \cdot \exp\left(\frac{-x}{\alpha \cdot iter_{\max}}\right) & R_2 < ST \\ M_{i,j}^t + Q \cdot L & R_2 \geq ST \end{cases} \tag{14}$$

where $t$ represents the iteration number; $X_{i,j}^t$ represents the $i$th sparrow in the $j$th dimension after $t$ iterations; $iter_{\max}$ is a constant representing the maximum number of iterations; $R_2$ and $ST$ are the alarm value and safety threshold, respectively; $\alpha$ is a random number between 0 and 1; $Q$ is a random number following normal distribution. $L$ is a matrix of size $1 \times y$ with all elements equal to 1.

The scroungers need to monitor the producers' predation. When producers have food, they will compete for it. If the scroungers fail, they will continue to monitor. The update of the scrounger's position is shown as follows:

$$M_{i,j}^{t+1} = \begin{cases} Q \cdot \exp\left(\frac{M_{worst}^t - M_{i,j}^t}{i^2}\right) & i > \frac{n}{2} \\ M_P^{t+1} + \left|M_{i,j}^t - M_P^{t+1}\right| \cdot A^+ \cdot L & i \leq \frac{n}{2} \end{cases} \tag{15}$$

where $A$ represents a matrix whose elements are either 1 or $-1$, and $A^+ = A^T\left(AA^T\right)^{-1}$; $M_p$ is the best position located by the producer; $M_{worst}$ represents the current worst position. When $i > n/2$, it means that $i$th scrounger has a risk of insufficient energy.

The initial position of sparrows in the population is randomly generated, the formula is shown below:

$$M_{i,j}^{t+1} = \begin{cases} M_{best}^t + \beta \cdot \left|M_{i,j}^t - M_{best}^t\right| & f_i > f_g \\ M_{i,j}^t + K \cdot \left(\frac{\left|M_{i,j}^t - M_{worst}^t\right|}{(f_i - f_w) + \varepsilon}\right) & f_i = f_g \end{cases} \tag{16}$$

where $M_{best}$ represents the current global optimal location; $\beta$ is the step control parameter; the value of $K$ is between $-1$ and 1; $f_i$ is the fitness value; $f_w$ and $f_g$ represent the current worst and best adaptation, respectively; $\varepsilon$ is the minimum constant. When $f_i > f_g$, this indicates that the sparrow is at the edge of the group; when $f_i = f_g$, this indicates that the sparrows in the middle of the population realize the danger and need to be closer to other sparrows.

### 3.4. The Proposed Fault Diagnosis Scheme

Based on RTSMRDE, t-SNE and SSA-SVM, the flowchart of proposed method is shown in Figure 5. The steps are summarized as follows:

First, the rotating machinery fault experimental platform collected the gearbox vibration signals under various failure conditions.

Second, the proposed RTSMRDE was used to compute the entropy values and create a high-dimensional feature set in order to completely extract the feature information of the gearbox fault signals.

Third, the dimensionality of the initial RTSMRDE feature set was reduced using the t-SNE to generate a sensitive low-dimensional feature set.

Finally, an optimized SVM model was constructed using SSA. The test set was fed into the optimized SVM for fault classification.

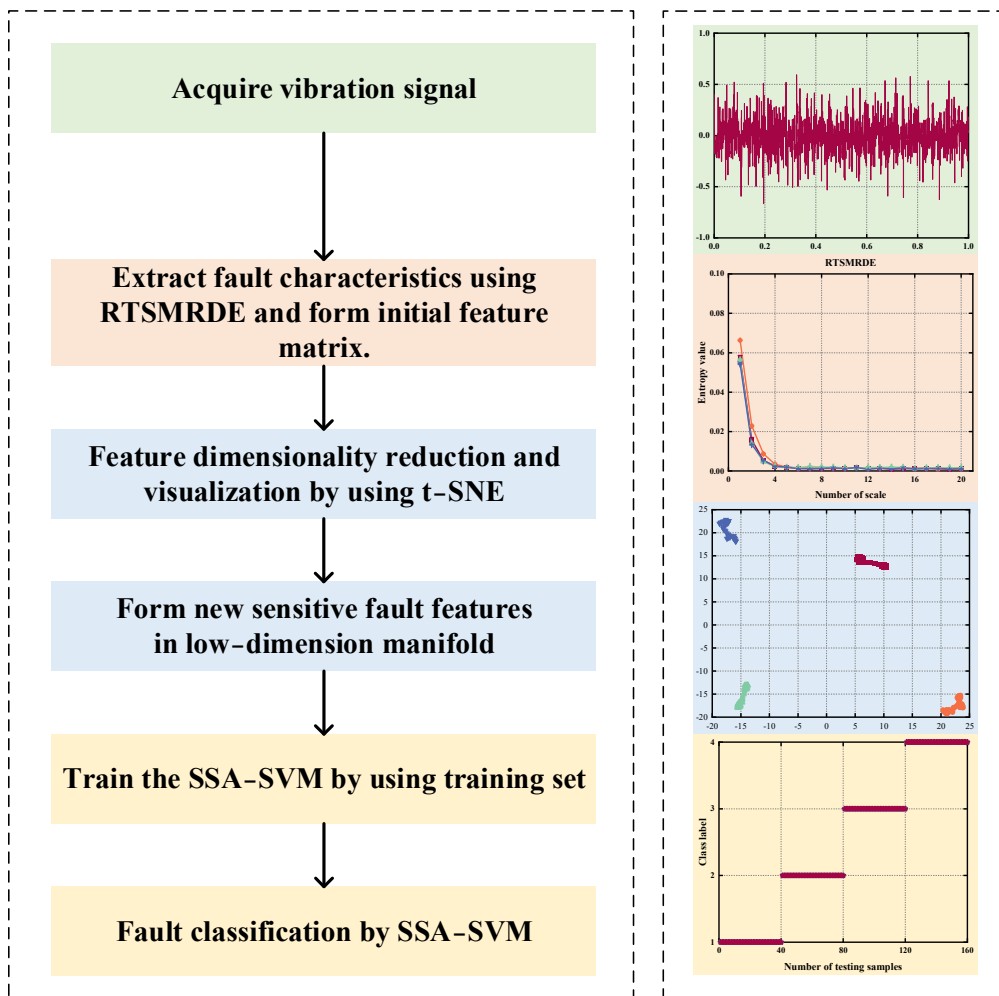

**Figure 5.** Flowchart of the fault diagnosis.

## 4. Experimental Verification

Through two gearbox fault experiments, the performance of the proposed fault diagnostic technique was evaluated. In addition, the proposed fault diagnosis model was compared with the existing diagnosis algorithm to verify its superior performance. All experiments were performed on the MATLAB R2022a environment running on an AMD Ryzen 7 5800 H 3.2 GHz, 16.0 GB RAM, and Windows 11 computer.

### 4.1. Case 1: Data from Southeast University Gearbox Dataset

4.1.1. Description and Division of Data

The experimental data (D1) of the gearbox in Case One were provided by Southeast University [42]. Dataset D1 selected the vibration signal collected by the Y-axis sensor on the planetary gearbox in the original dataset, which was operated under the condition that the load of the speed system was set to 20 Hz–0 V. As shown in Table 5, the D1 dataset had five operating states, namely Normal, Chipped tooth, Surface wear, Root wear, and Missing tooth. Each operating state consisted of 50 sub-samples with a data length of 2048. Therefore, there was a total of 250 samples in dataset D1. These samples were divided into 10 training samples and 40 testing samples for fault diagnosis. The time-domain signals of the five operating states are shown in Figure 6, where it can be clearly seen that there were obvious impact components in the time-domain waveform under faulty conditions.

**Table 5.** Description of D1.

| Fault Types | Motor Speed (r/min) | Number of Training Samples | Number of Testing Samples | Class Label |
|---|---|---|---|---|
| Normal | 1200 | 10 | 40 | NOR |
| Chipped tooth | 1200 | 10 | 40 | CTF |
| Surface wear | 1200 | 10 | 40 | SWF |
| Root wear | 1200 | 10 | 40 | RWF |
| Missing tooth | 1200 | 10 | 40 | MTF |

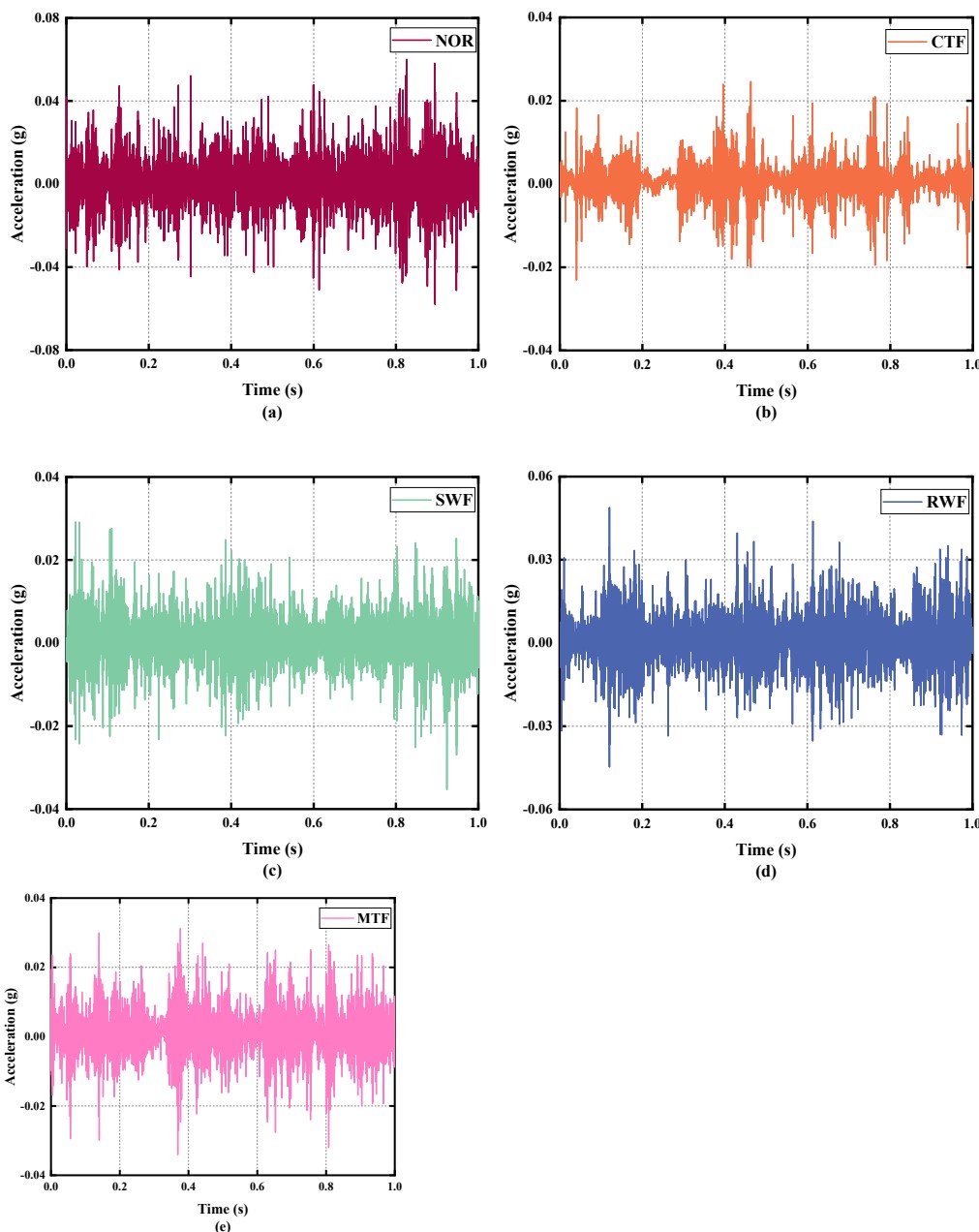

**Figure 6.** The time-domain signals of D1 in the experiment. (**a**) NOR; (**b**) CTF; (**c**) SWF; (**d**) RWF; (**e**) MTF.

4.1.2. Feature Extraction for D1

Using the RTSMRDE as a feature extraction tool, fault features were extracted from the experimental dataset D1 consisting of 250 samples. In the end, we obtained a $250 \times 20$ fault feature matrix, where 250 is the number of samples and 20 is the dimension

of feature extraction. Figure 7 shows the error bar plot of entropy calculation for dataset D1. It can be seen from the figure that the trends of the five fault states in the multiscale calculation were the same and stable; except for a small error in the normal state, the errors in the other states were almost invisible. This indicates that the RTSMRDE algorithm can effectively overcome the defects of entropy calculation in multiscale and control the entropy bias phenomenon.

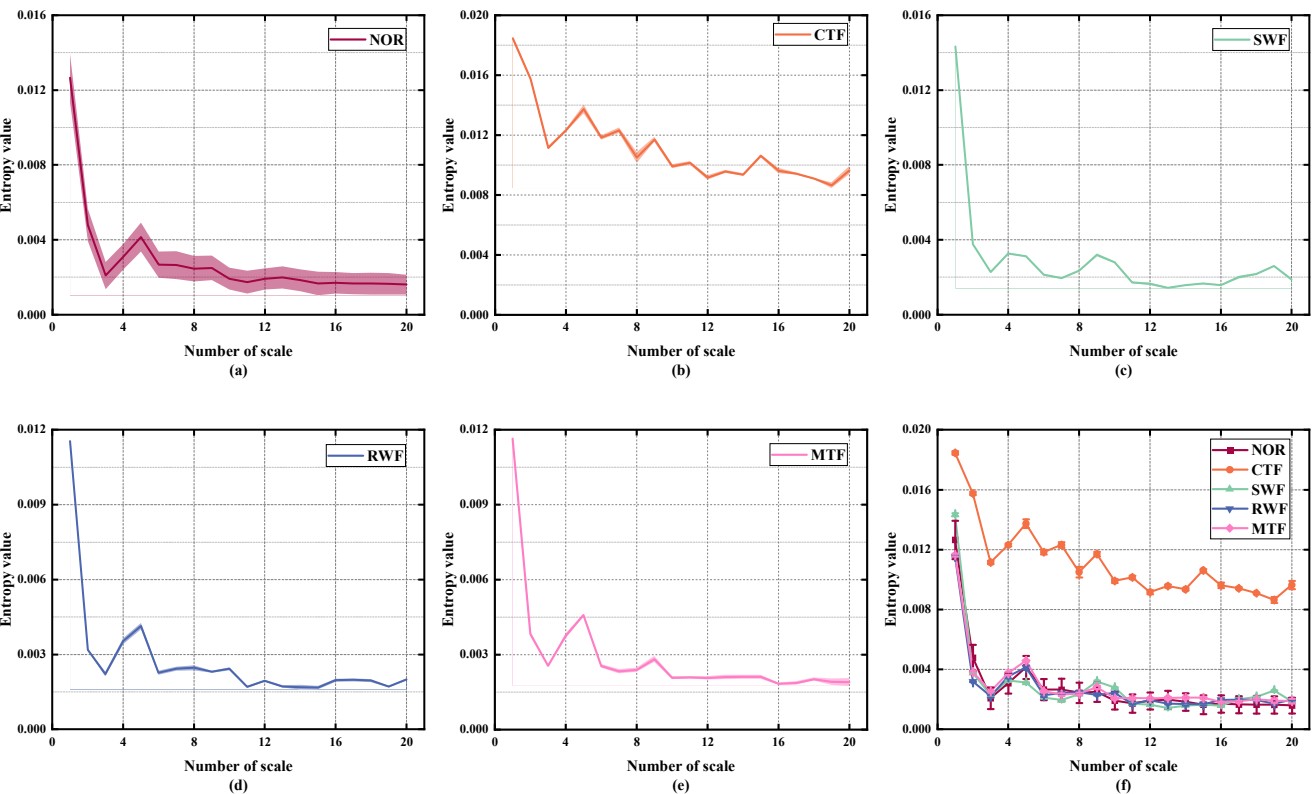

**Figure 7.** RTSMRDE value on D1. (**a**) NOR; (**b**) CTF; (**c**) SWF; (**d**) RWF; (**e**) MTF; (**f**) Five kinds of fault entropy values.

### 4.1.3. Data Reduction and Visualization

Relying solely on the entropy curve to determine the fault status of the gearbox is a challenging task. Therefore, the t-SNE dimension reduction algorithm was employed to process the high-dimensional feature set. The reduced fault feature set only consisted of the most sensitive fault features, which not only saved time in computation but also facilitated visualization. As shown in the Figure 8, the visual results of the sample features obtained after two-dimensional and three-dimensional dimension reduction show that each group of samples was well-clustered, with no sample mixing or blurred boundaries observed among the five operating status samples of the gearbox. These results indicate that the t-SNE algorithm is capable of effectively extracting crucial information from high-dimensional features and accurately identifying the status of the gearbox.

### 4.1.4. Analysis of Diagnosis Results

Finally, the intelligent diagnosis of the gearbox faults was achieved by inputting the low-dimensional sensitive feature set into SSA-SVM. To train the model, 10 sets of samples for each state were randomly selected as the training set, and the remaining 40 sets were used as the test set. There was a total of 50 training samples and 200 test samples. The SSA was used to optimize the SVM parameters. Then, the optimized support vector machine model was established, and the test set was input into the model for classification. As

shown in the Figure 9, the proposed method could effectively identify various faults with a recognition accuracy of up to 100%.

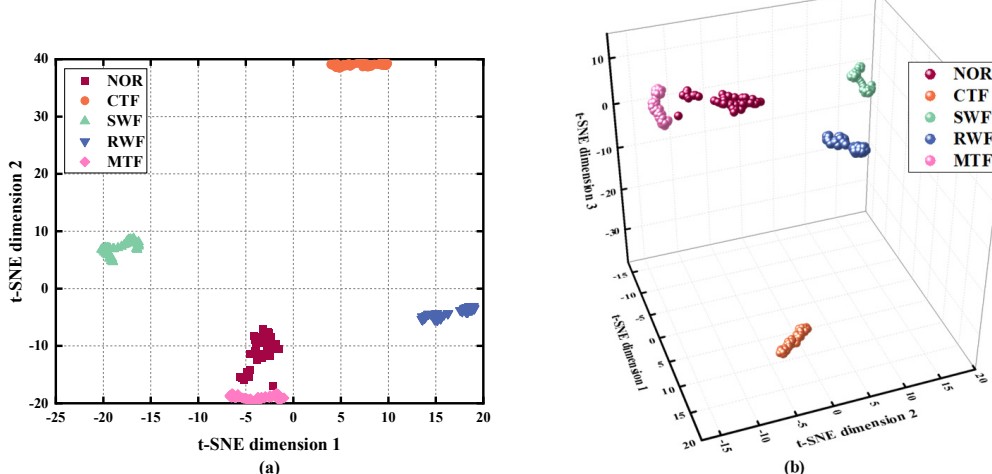

**Figure 8.** (**a**) The 2D result obtained using the t-SNE; (**b**) The 3D result obtained using the t-SNE.

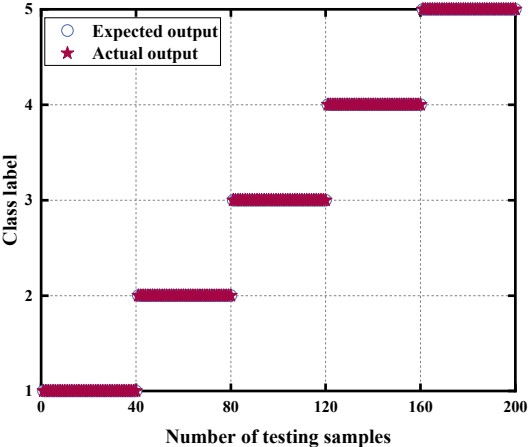

**Figure 9.** Fault classification of RTSMRDE on D1.

*4.2. Case 2: Data from MFS*

4.2.1. Description and Division of Data

The gearbox data set (D2) was provided by a mechanical failure simulation experiment system (MFS). The MFS manufactured by SQI company can simulate various common mechanical equipment, and the modular component design of the experimental bench is powerful and reliable, so it can be used to simulate common bearing and gear failures of the wind turbine drive train. As shown in Figure 10, the main part of the experimental system consisted of the integrated mechanical fault simulation experimental bench and the data acquisition equipment. As shown in Figure 11, the gearbox fault diagnosis study kit used for the experiment consisted of a missing tooth fault gear, a broken tooth fault gear, and a tooth wear gearbox.

The experiments were conducted at a motor speed of 1750 rpm to test the normal state and three fault states, respectively. The original samples were divided into 50 sub-samples, and each sample was 2048 in length. The D1 is shown in Table 6. The vibration signals of four faults are shown in Figure 12.

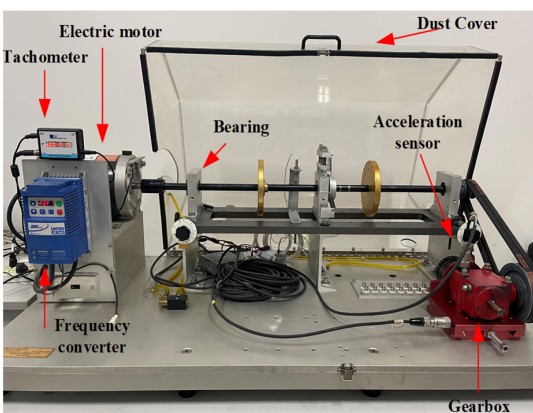

**Figure 10.** Comprehensive mechanical failure simulation experiment system.

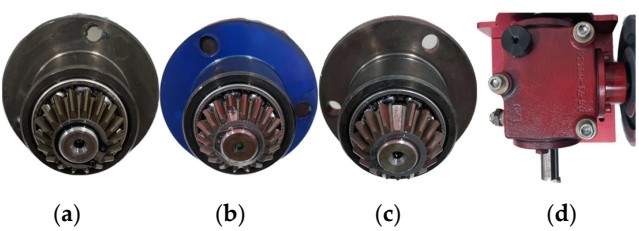

(**a**)          (**b**)          (**c**)          (**d**)

**Figure 11.** Four gear health states: (**a**) normal condition; (**b**) broken tooth fault gear; (**c**) missing tooth fault gear; (**d**) the wear tooth gearbox.

**Table 6.** Description of D2.

| Fault Types | Motor Speed (r/min) | Number of Training Samples | Number of Testing Samples | Class Label |
|---|---|---|---|---|
| Normal | 1750 | 10 | 40 | NOR |
| Broken tooth | 1750 | 10 | 40 | BTF |
| Missing tooth | 1750 | 10 | 40 | MTF |
| Wear tooth | 1750 | 10 | 40 | WTF |

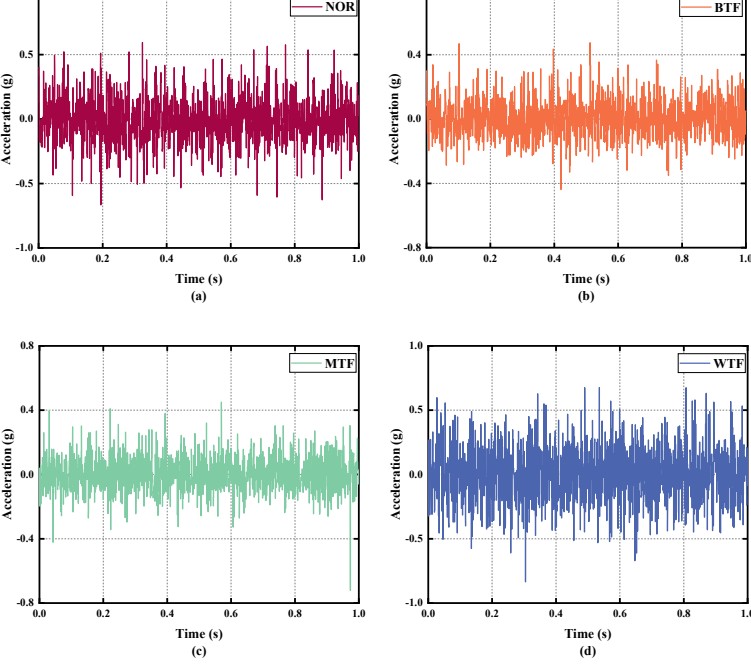

**Figure 12.** The time-domain signals of D2. (**a**) NOR; (**b**) BTF; (**c**) MTF; (**d**) WTF.

### 4.2.2. Feature Extraction for D2

The MFS experimental platform collected vibration signals from four faults and selected 50 sample signals for each fault. The entropy value was calculated for these 200 samples, and the RTSMRDE algorithm was used to extract feature from the fault signals. The resulting fault feature matrix had a size of 200 × 20, where 200 represents the number of samples and 20 represents the number of dimensions. As shown in the Figure 13, the RTSMRDE values indicate a stable trend, and the error value was extremely small, making it nearly invisible in the error bar graph. Zooming in on the region of scale range from 4 to 20 in Figure 13e, it can be seen that the entropy values calculated for the four fault states were not equal but existed in an alternating manner. Although using RTSMRDE can make the results of entropy calculation more stable and reduce entropy bias, directly using a high-dimensional feature matrix for classification cannot produce the best results.

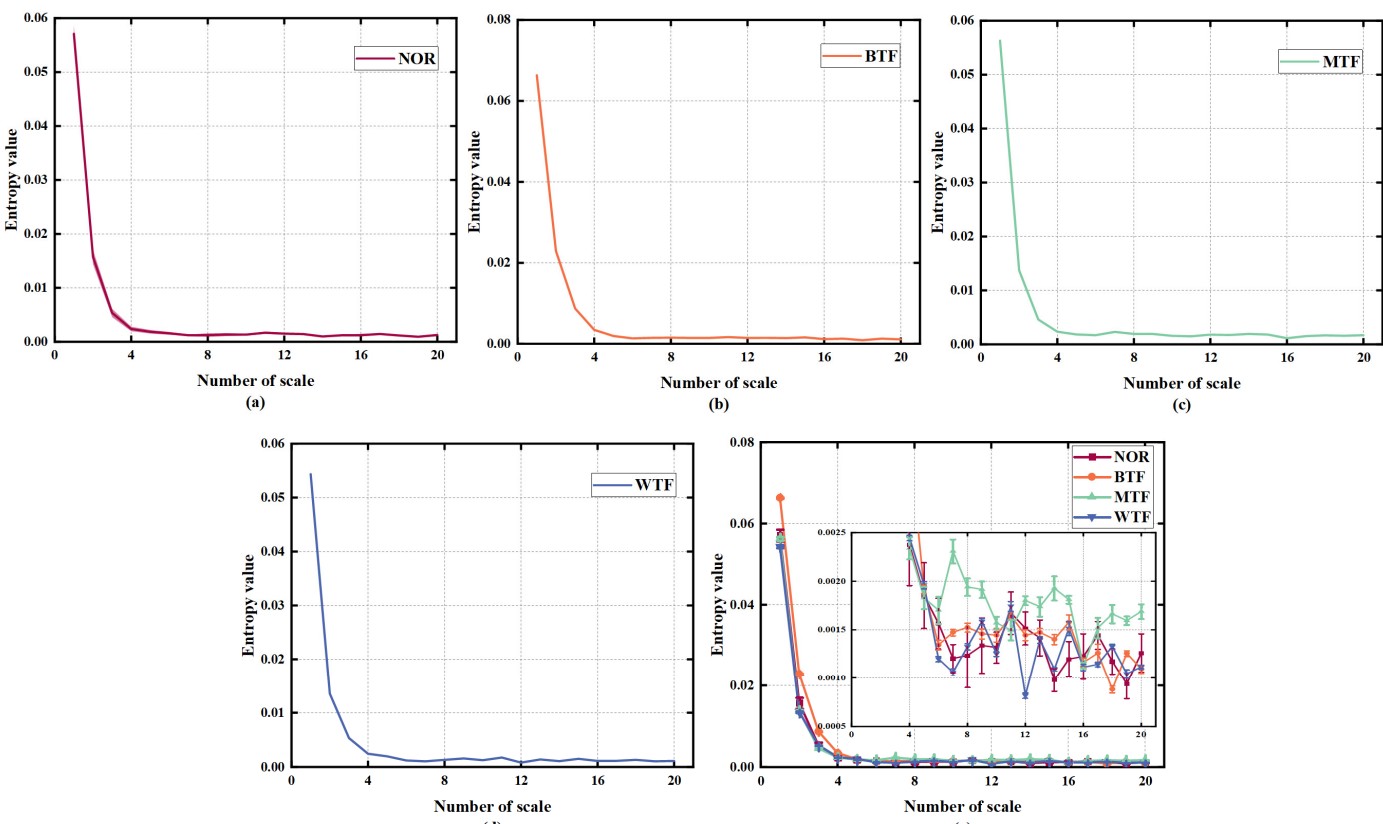

**Figure 13.** RTSMRDE value on D2. (**a**) NOR; (**b**) BTF; (**c**) MTF; (**d**) WTF; (**e**) Four kinds of fault entropy values.

### 4.2.3. Data Reduction and Visualization

Using the t-SNE algorithm to reduce the size of the initial feature matrix of 200 × 20, the visualization results of the dimensionality reduction are shown in Figure 14, and the four fault states are clearly distinguished. Each fault state signal existed independently in different regions without confusion. The 2D and 3D visualization results both showed ideal results. This indicates that the t-SNE algorithm can effectively extract key information from high-dimensional features, making it easier to distinguish the state of the gearbox.

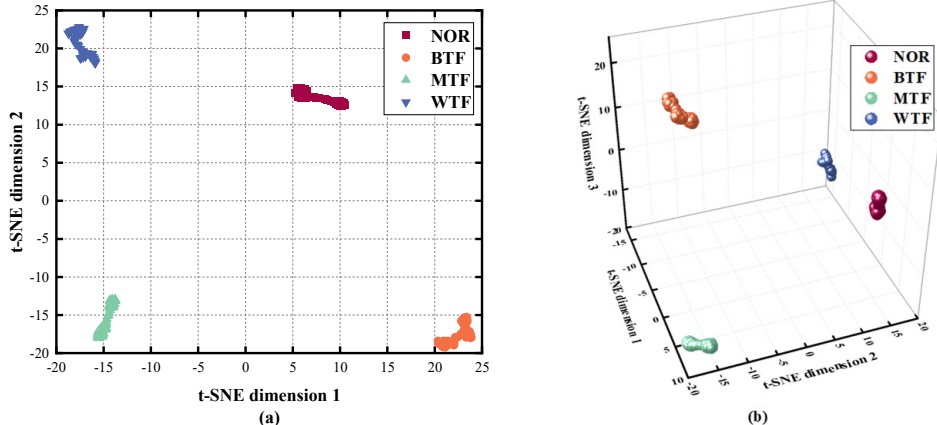

**Figure 14.** (**a**) The 2D result obtained using the t-SNE; (**b**) The 3D result obtained using the t-SNE.

### 4.2.4. Analysis of Diagnosis Results

The fault classification was achieved using SSA-SVM. As shown in the Figure 15, the method proposed in this paper could effectively identify various faults with a recognition accuracy of up to 100%.

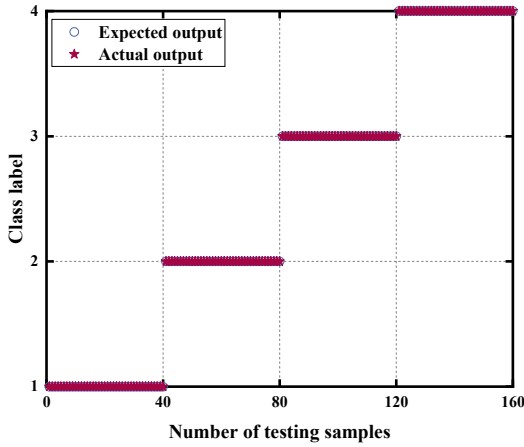

**Figure 15.** Fault classification of RTSMRDE on D2.

### *4.3. Contrast Analysis*

To fully verify the advantages of the proposed fault diagnosis model, we used two sets of control experiments to compare the advantages of RTSMRDE and data dimension reduction in the fault diagnosis model.

### 4.3.1. Comparison of RTSMRDE with Other Different Entropy Algorithms

To verify the advantages of the RTSMRDE, it was compared with six other entropy algorithms mentioned in Table 3 using the D1 and D2 dataset. The entropy calculation results of each algorithm are shown in Figures 16 and 17, respectively. The entropy values calculated by the MSE, MFE, MDE, and MRDE algorithms showed increasing errors with increasing scale, and the entropy curves showed significant fluctuations. However, the refined composite entropy algorithm showed smaller errors and smoother curves, indicating that the algorithm is more stable and effective. The entropy curve calculated by the samples processed by the refined time-shift algorithm showed the smallest error and the most stable curve among all the results, indicating the best performance.

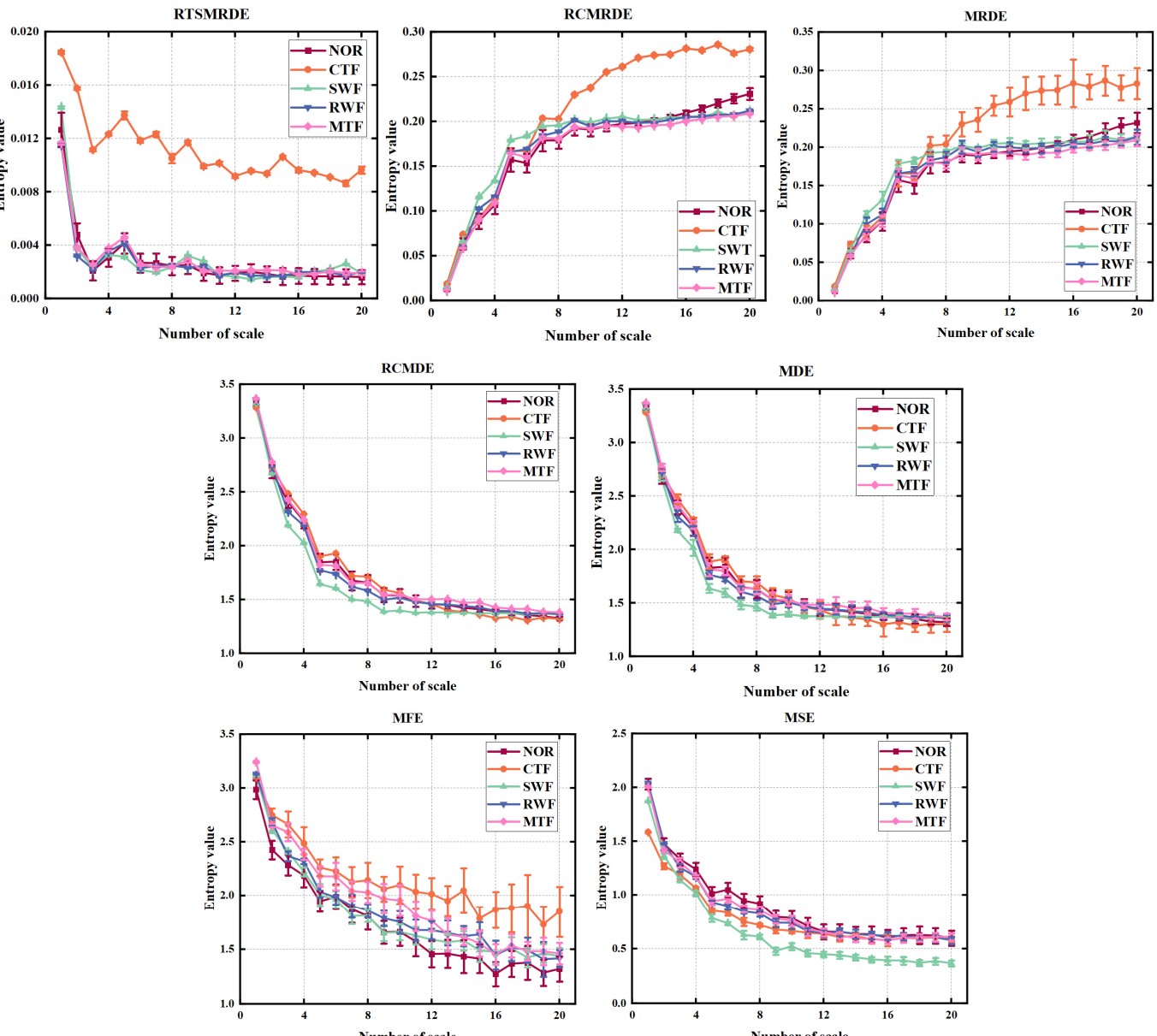

**Figure 16.** Different entropy values of D1.

The feature sets extracted by each algorithm were sent to the SSA-SVM for classification, and the results are shown in Figures 18 and 19, respectively. As shown in Figure 18, the numbers of the misclassified samples for RTSMRDE, RCMRDE, MRDE, RCMDE, MDE, MFE, and MSE on D1 were 1, 5, 7, 5, 8, 11, and 15, respectively. The average correct rate of the RTSMRDE algorithm was 99.5%, which was the highest among the seven algorithms, with accuracy rates of 2%, 3%, 2%, 3.5%, 5.5%, and 7.5% higher than the other six algorithms, respectively. As shown in the Figure 19, the numbers of the misclassified samples for RTSMRDE, RCMRDE, MRDE, RCMDE, MDE, MFE, and MSE on D2 were 2, 3, 7, 2, 8, 10, and 13, respectively. The RTSMRDE algorithm exhibited an average correct rate of 98.75%, surpassing the other six algorithms by 0.625%, 3.125%, 0%, 3.75%, 5.625%, and 6.875% in terms of accuracy rates.

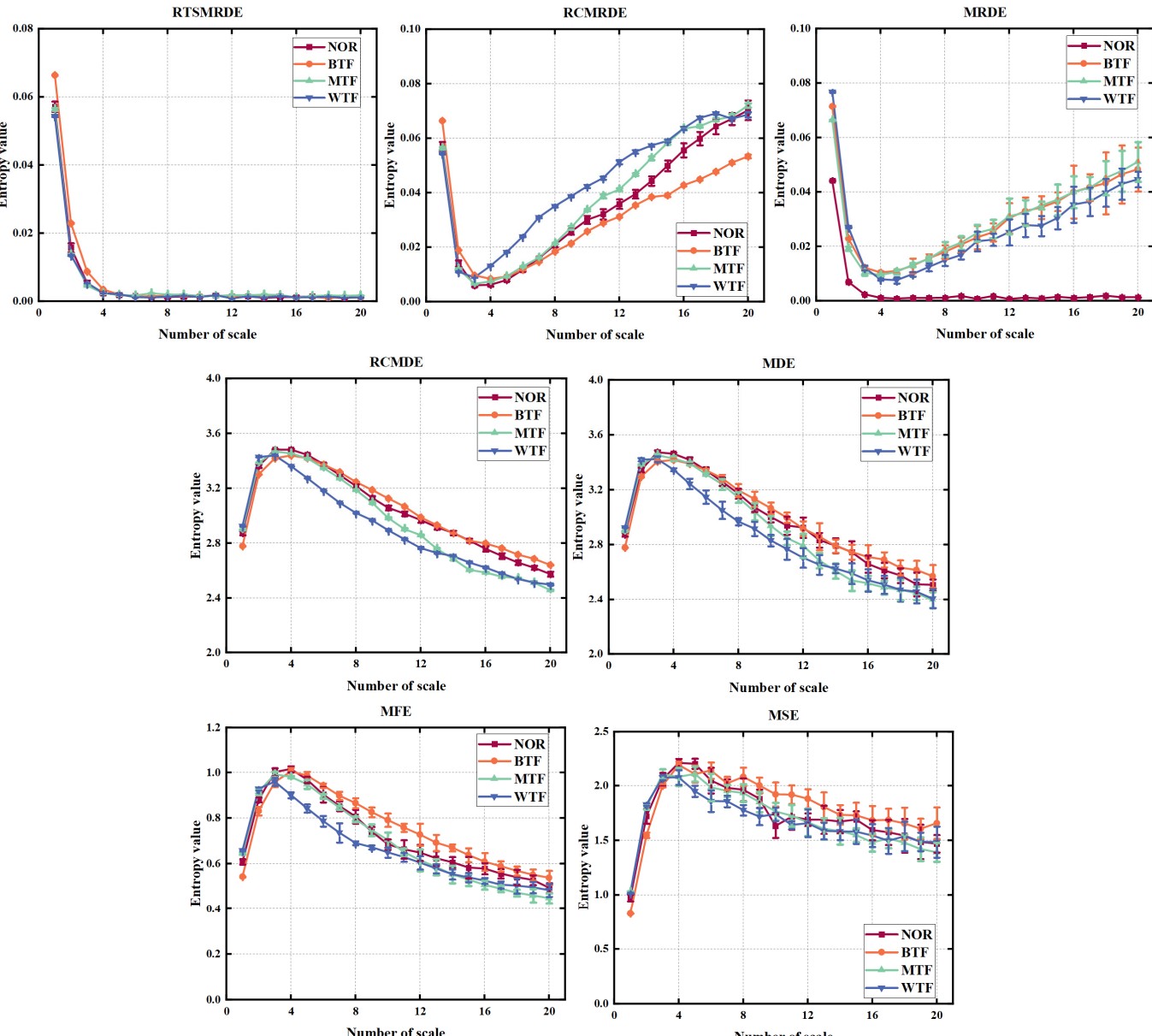

**Figure 17.** Different entropy values of D2.

Figure 20 shows the accuracy of different algorithms on two datasets. In addition, the running time of each algorithm is shown in Table 7. It is clear that the MDE and MRDE algorithms had the shortest computation time of approximately 2 s. The RTSMRDE algorithm took approximately 4 s, which was the second shortest. The computation time of other algorithms exceeded 10 s, which was three to four times longer than that of RTSMRDE. In summary, the proposed method performed better for fault classification of the gearbox dataset.

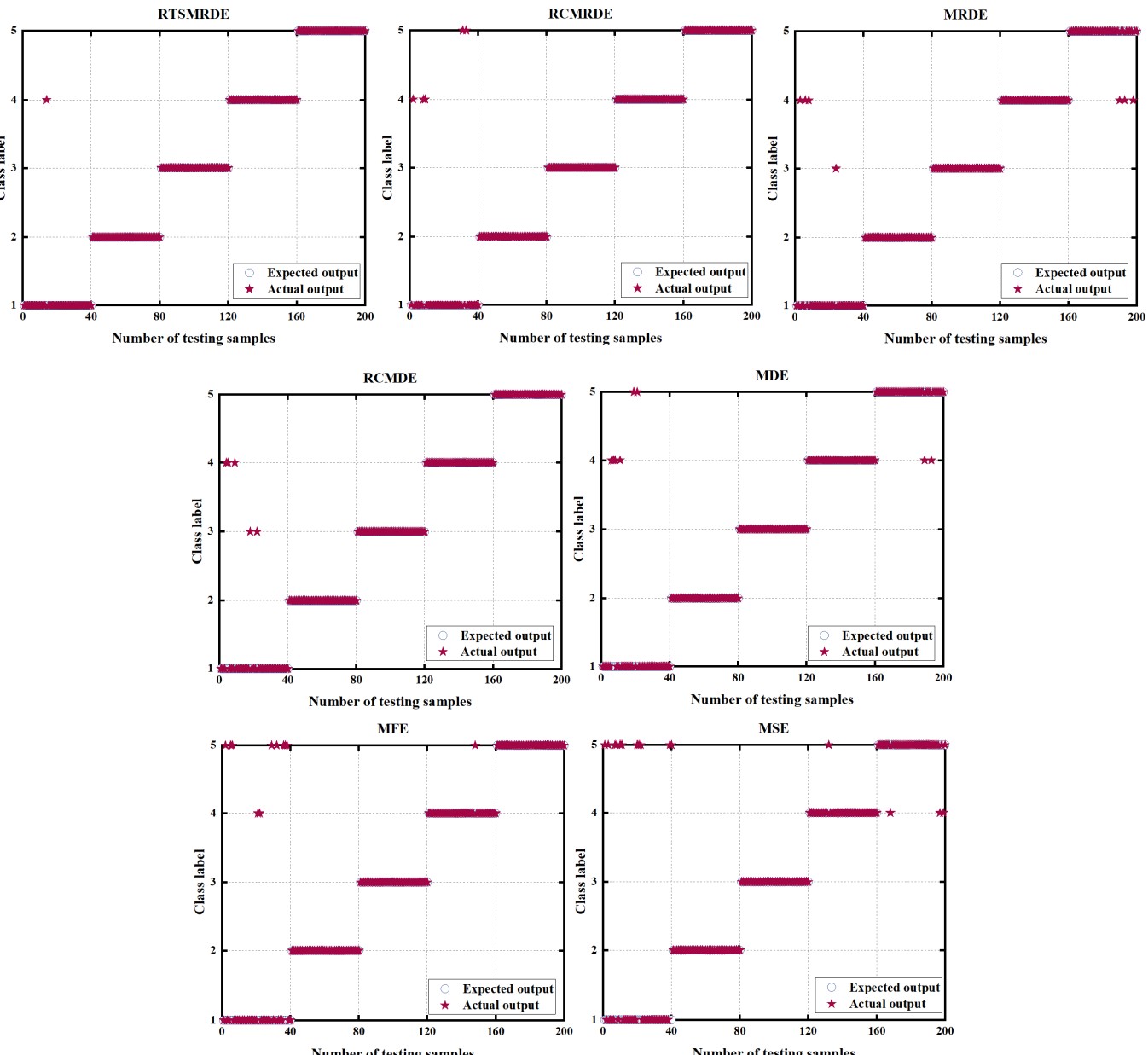

**Figure 18.** The fault diagnosis result of D1.

### 4.3.2. Comparison between Using and Not Using Data Reduction Methods

By comparing Figures 9 and 15, the first subfigures of Figure 17, and the first subfigure of Figure 18, it can be observed that the classification result obtained without using data dimension reduction algorithm had one misclassified sample in D1 and two misclassified samples in D2. However, after applying the data dimension reduction algorithm, all samples could be correctly classified. This indicates that the sensitive feature set extracted through data reduction can effectively improve the classification accuracy.

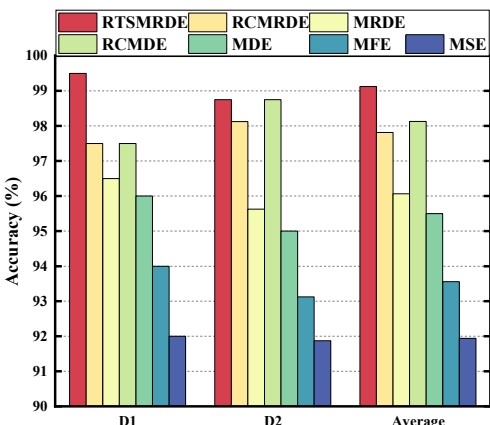

**Figure 19.** The fault diagnosis result of D2.

**Figure 20.** Accuracy of different entropy methods.

**Table 7.** Running times for different entropies.

| Data | RTSMRDE | RCMRDE | MRDE | RCMDE | MDE | MFE | MSE |
|------|---------|--------|------|-------|-----|-----|-----|
| D1 | 4.62 s | 17.56 s | 2.40 s | 17.65 s | 2.36 s | 19.26 s | 14.08 s |
| D2 | 3.76 s | 14.23 s | 1.81 s | 14.36 s | 1.83 s | 15.77 s | 12.22 s |

## 5. Conclusions

The vibration signal of a gearbox has non-continuous and non-linear characteristics. When a gearbox local fault occurs, periodic fault signals with impact characteristics will appear. Therefore, we proposed a feature extraction method based on RTSMRDE, t-SNE, and SSA-SVM to construct a new intelligent diagnosis method for gearbox faults. The effectiveness and superiority of this method compared to existing methods were verified through simulation signals and fault simulation experiments. The proposed method had a positive impact on the fault diagnosis of rotating components, such as gearboxes in wind turbines. The following conclusions have been drawn:

1. The RTSMRDE is based on MRDE, combined with the ideas of time shifting coarse-graining operations. It overcomes the shortcomings of traditional multiscale reverse dispersion entropy and can effectively and comprehensively extract the fault characteristics of gearboxes.
2. The t-SNE can effectively remove redundant features in high-dimensional fault feature sets, thus obtaining a sensitive and easily classifiable low-dimensional feature set.
3. Constructing a novel diagnosis model for gearbox faults based on RTSMRDE, t-SNE, and SSA-SVM.
4. The proposed method was validated with noise signals and experimental datasets and demonstrated a more prominent overall performance in terms of feature extraction capability and computational speed.

**Author Contributions:** Conceptualization, X.W. and H.J.; methodology, X.W. and H.J.; validation, X.W. and H.J.; investigation, X.W. and H.J.; resources, X.W.; data curation, X.W. and H.J.; writing—original draft preparation, H.J.; writing—review and editing, X.W.; visualization, H.J.; supervision, X.W.; project administration, X.W.; funding acquisition, X.W. All authors have read and agreed to the published version of the manuscript.

**Funding:** The project of software and hardware development of online vibration condition monitoring and intelligent fault diagnosis for rotating machinery (3612403222440) supported by scientific research Foundation of Nanjing Institute of Technology.

**Data Availability Statement:** The data presented in this study are available on request from the corresponding author.

**Conflicts of Interest:** The authors declare no conflict of interests.

## Abbreviations

The following abbreviations are used in this manuscript:

| | |
|---|---|
| MSE | Multiscale Sample Entropy |
| MFE | Multiscale Fuzzy Entropy |
| MPE | Multiscale Permutation Entropy |
| DE | Dispersion Entropy |
| MDE | Multiscale Dispersion Entropy |
| RCMDE | Refined Composite Multiscale Dispersion Entropy |
| RDE | Reverse Dispersion Entropy |
| MRDE | Multiscale Reverse Dispersion Entropy |
| RCMRDE | Refined Composite Multiscale Reverse Dispersion Entropy |
| RTSMRDE | Refined Time-Shifted Multiscale Reverse Dispersion Entropy |

| t-SNE | t-distributed Stochastic Neighbour Embedding |
| SSA-SVM | Sparrow Search Algorithm-Support Vector Machine |
| PN | pink noise |
| WN | white noise |

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
