# Peer review of "Gearbox Fault Diagnosis Based on Refined Time-Shift Multiscale Reverse Dispersion Entropy and Optimised Support Vector Machine"

_machines, doi:10.3390/machines11060646_

Round 1
Reviewer 1 Report
The research design of this paper is sound, the information flow of the paper is well organized, and the conclusion is relevant.
The structure of the research paper is good. But I do have some questions about its details.
One question I would like to bring out is the fault features. In this paper, it addresses how high-dimensional feature set was created from vibration signals and how feature dimensionality was reduced. On page 13, "20 is the dimension of feature extraction" On page 16, "200 represents the number of samples and 20 represents the number of dimensions". But the paper did not list these high- or low-dimensional features, which should be important. Without knowing the features other researcher cannot repeat and test their research. There is no need to explain how these features were extracted in detail, but should at least list the name of these features.
Maybe I overlooked or did not pay close attention, the paper did not specifically explain how these features are linked, and inputted to the mathematical model for computing entropy value. If the paper has already include this, then it should be enhanced and strengthened. This part I am not very sure. But would like to share my thoughts.
Secondly, I would suggest to include references to support the selection of parameters in the model on Page 5.
Thirdly, the paper mentioned white noise and pink noise. It would be better if the authors added one sentence to explain what the terms mean in machine fault detection to distinguish from the medical perspective.
The workload of this research is remarkable. Very impressive.
Author Response
Dear Reviewer,
Thank you very much for your recognition of our work and for the comments you provided in the peer review process. Regarding your comments, we provide the following responses:
Firstly, the feature dimensions mentioned in this paper are essentially a type of values obtained when calculating the multi-scale entropy of a signal at different scales. For example, when calculating the multi-scale entropy of a signal, if we set the scale to 20, it means that the signal will be computed for 20 entropy values at scales 1 to 20, each of which represents a feature value of the signal at the corresponding scale. Each entropy value is equivalent to a manifestation of a feature in a certain dimension. Here, the dimension is not like the dimensions in traditional feature extraction in time and frequency domains, so we cannot provide names such as mean, standard deviation, etc.
Secondly, in the second paragraph of Section 2.4 on parameter selection, we have already marked the reference used for parameter selection.
Thirdly, inspired by your suggestion, we have added an explanation of white noise and pink noise in the context of mechanical fault detection in the first paragraph of Section 2.4. We believe that this addition will improve the readability of this section.
Once again, we sincerely appreciate your valuable comments on our article. These comments will help us to better complete our work.
Best regards,
Han Jiang et al.
Reviewer 2 Report
This paper proposes a gearbox fault diagnosis method based on Refined Time-Shifted Multiscale Reverse Dispersion Entropy (RTSMRDE), t-dis-tributed Stochastic Neighbor Embedding (t-SNE), and Sparrow Search Algorithm Support Vector Machine (SSA-SVM). The proposed RTSMRDE is used to calculate multiscale fault features. By incorporating the refined time-shift method into the Multiscale Reverse Dispersion Entropy, errors that arise during the processing of complex time series can be effectively reduced. Then t-SNE algorithm is utilized to extract sensitive features from the multiscale, high-di-mensional fault features.
However, the paper is well written I have some remarks before accepted this work
· The abstract must be rewritten to show clearly the new results only.
· The authors used many abbreviations and acronyms, so I advise them to insert a table to include all used abbreviations
· The mathematical relations from 1 – 16 must be constructed or cite its references
· The "problem description" is not clear, overall, the methodology is not rigorously derived and explained. The derivation procedure is sometimes hard to follow, and a more thorough explanation of the passages should be provided
· The translation of figures should be included properly
· For the "Conclusions". I would suggest including a paragraph explaining the proposed model's novelty, the advantages and limitations of the approach, and possible practical applications.
· The authors have to shed light on the similarities and differences among their work and the literatures of the problem. A clear explanation, what is the new result in their work, and how it is build up upon previous work in the field.
I think that the authors should scan the whole paper to avoid the grammatical mistakes.
Author Response
Dear Reviewer,
Thank you very much for your recognition of our work and for the comments you provided in the peer review process. Regarding your comments, we provide the following responses:
Firstly, we have carefully reviewed the content of the abstract and made modifications to highlight the results of the article. We hope that the revised abstract can more accurately express the main idea of this article.
Secondly, thank you for your suggestion, we have added an abbreviation table at the end of the article.
Thirdly, in the first paragraph of Sections 2.1 and 3.3, we have provided reference citations for the relevant formulas used.
Fourthly, all theoretical processes have been strictly derived. If there are still unclear statements, we hope you can point out the deficiencies.
Fifthly, we have carefully checked all the pictures and their corresponding descriptions, and adjusted the layout of the article to make the text closer to the mentioned pictures. We hope that these modifications can increase the readability of the article.
Sixthly, inspired by your suggestion, we have added a description of practical applications and future work in the conclusion. We hope to increase the readability and completeness of the article.
Seventhly, in the introduction, we elaborated on the development process of entropy algorithms in feature extraction, as well as the drawbacks of traditional coarse-graining processing of time series. We emphasized that, in response to these issues, we used a new coarse-graining method based on multiscale reverse dispersion entropy to transform the feature extraction method, and demonstrated the effectiveness of the method through experiments. Finally, we outlined the innovative content and outstanding contributions of the article in the conclusion.
Once again, we appreciate your valuable comments on our paper. These comments will help us to better complete our work.
Best regards,
Han Jiang et al.
Reviewer 3 Report
In this work titled Gearbox fault diagnosis based on Refined Time-Shift Multiscale Reverse Dispersion Entropy and Optimised Support Vector Machin , the authors dealt with data reduction; fault diagnosis; gearbox; reverse dispersion entropy; support vector ma[1]chine.
The topics are presented into: Introduction, Refined Time-Shift Multiscale Reverse Dispersion Entropy, Experimental Verification, . Conclusions and 42 references
The present study demonstrates that the proposed method has been validated with noise signals and experimental datasets, and it had demonstrated a more prominent overall performance in terms of feature extraction capability and computational speed, according to the authors
The paper is well written and with no mistakes and the authors addressed the main question posed .The arguments presented by the authors were consistent with the evidence and arguments presented by the authors . The paper is original . The concluding remarks were supported by the data,
In my opinion , the authors must include the influence of parametric errors on the proposed approach could be also included
In this work titled Gearbox fault diagnosis based on Refined Time-Shift Multiscale Reverse Dispersion Entropy and Optimised Support Vector Machin , the authors dealt with data reduction; fault diagnosis; gearbox; reverse dispersion entropy; support vector ma[1]chine.
The topics are presented into: Introduction, Refined Time-Shift Multiscale Reverse Dispersion Entropy, Experimental Verification, . Conclusions and 42 references
The present study demonstrates that the proposed method has been validated with noise signals and experimental datasets, and it had demonstrated a more prominent overall performance in terms of feature extraction capability and computational speed, according to the authors
The paper is well written and with no mistakes and the authors addressed the main question posed .The arguments presented by the authors were consistent with the evidence and arguments presented by the authors . The paper is original . The concluding remarks were supported by the data,
In my opinion , the authors must include the influence of parametric errors on the proposed approach could be also included
Author Response
Dear Reviewer,
Thank you very much for your recognition of our work and for the comments you provided in the peer review process. Regarding your comments, we provide the following responses:
Firstly, regarding your concerns about the impact of parameter errors, we have carefully considered this issue. In Section 2.4, we have conducted experiments to help us determine the optimal parameters for the experiment, and we have analyzed the results of using different values for different parameter selections. We hope to use this approach to select the best parameters to achieve the best performance of the model. Moreover, the parameters used for comparison with other models in the paper are taken from corresponding papers and have been validated in those papers.
Secondly, regarding the quality of the English language, you found it very difficult to understand. We apologize for this and also feel confused. We have reviewed the content of the paper, and made modifications to some errors. If there are still areas that are difficult to understand, please let us know.
Once again, we appreciate your valuable comments on our paper. These comments will help us to better complete our work.
Best regards,
Han Jiang et al.
Reviewer 4 Report
In general, the paper under review makes a positive impression: the results obtained demonstrate the advantages of the proposed RTSMRDE method over the rival techniques for the classsification of gearbox failures, which is also confirmed by the authors in the experimental setup.
The minor notes are following.
Sinse the authors use the runtime as an indicator of the performance for the algorithm comparison, it is necessary to provide information about the configuration of the computer utilized.
The manuscript needs to be edited to address formating issues. This applies to the design of the abstract, different fonts on page 5, missing numbers in the references to the table on page 6, and figure on page 16. For the convenience of a potential reader, I may also recommend changing the manuscript design so that the graphic content would be closer to the text in which it is mentioned.
Minor proofread aimed at spelling, articles usage, and punctuation is required.
Author Response
Dear Reviewer,
Thank you very much for your recognition of our work and for the comments you provided in the peer review process. Regarding your comments, we provide the following responses:
Firstly, thank you for the reminder, and we have added information on the computer configuration in the experimental validation section of Chapter 4. We believe that this will be an important reference indicator for other researchers.
Secondly, we have rearranged the position of the figures and tables in the paper to make the graphics closer to the mentioned text and increase the readability of the article.
Thirdly, we have carefully checked and corrected any spelling and punctuation errors in the article.
Once again, we appreciate your valuable comments on our paper. These comments will help us to better complete our work.
Best regards,
Han Jiang et al.